# REFORMER: A DEEP LEARNING MODEL FOR RUN-TIME SELECTION OF CONVOLUTION KERNELS

## ABSTRACT

As neural networks grow larger, optimizing GPU kernel selection becomes increasingly essential to minimizing the time, cost, and energy demands of model training and inference. Current methods rely on hand-written rules-based heuristics, which often yield suboptimal performance, are labor-intensive to develop, and are difficult to adapt across hardware architectures and firmware releases. In this paper, we frame kernel selection as a sequence classification problem solved on the CPU, thereby leaving GPU resources free for user training and inference tasks. Traditional transformers are less effective in this context because CPU deployment limits the advantages of parallelism in attention mechanisms. In this regard, we propose the *Γ-block*, which performs only three matmul operations compared to the six required by a transformer block, while maintaining the same depth in terms of learnable layers. Our experiments on the IMDB and Reuters datasets demonstrate that a small model based on the Γ-block delivers comparable sequence classification accuracy to a similar model based on transformer blocks, while also providing faster inference times on the CPU. By stacking multiple Γ-blocks, we develop a lightweight model for kernel selection, named *Reformer*. To train the model, we propose a novel approach that assigns optimality probabilities to kernels based on their runtimes, offering a more robust alternative to one-hot probabilities. We demonstrate the effectiveness of Reformer by integrating it into MIOpen for convolution kernel selection, achieving an average speed-up of approximately 3x in convolution operations on the AMD Instinct™ MI100 GPU.

## 1 INTRODUCTION

Over the last decade, neural network sizes and data set sizes have had a reinforcing relationship: as more data has become available, models have been enlarged to better learn from it, and concurrently, larger models have driven the collection of more data to fully harness their potential. This feedback loop has led to exponential growth in model sizes over time (Figure 1), resulting in a significant increase in the compute and energy requirements (Strubell et al., 2020; Patterson et al., 2022; Dodge et al., 2022) for training and deploying these models. Although researchers have worked to scale down neural networks to reduce their compute footprint (Hinton et al., 2015; Cheng et al., 2017; Micikevicius et al., 2017), recent findings suggest that large neural networks are not merely a passing trend but a key element of high-performing machine learning systems (Bubeck & Sellke, 2021).

To meet the increasing compute demands of large models, researchers from both academia and industry have been actively developing solutions. On the hardware front, these efforts have led to improvements in GPU architectures (AMD, 2021; Nvidia, 2020) and the introduction of advanced AI accelerators (Jouppi et al., 2023; Keller et al., 2022). On the software side, research endeavors have focused on developing computationally-efficient, hardware-optimized kernels[1] for common machine learning operations. However, since these kernels are hardware-specific and their performance varies across the problem space in a highly discontinuous and non-linear fashion, choosing the optimal kernel for any given problem is a non-trivial task. Brute-force approaches are generally impractical in this regard. Exhaustively searching the problem space and benchmarking all applicable kernels to determine the best one leads to combinatorial complexity, determined by the

---

[1]In this article, the word "kernel" refers to a *compute kernel* (also known as a *GPU kernel*). To avoid ambiguity, we will refrain from using the term "kernel" to refer to the filter of a convolution operation.

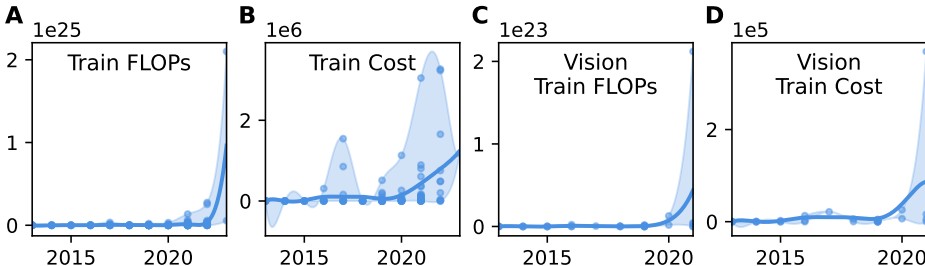

Figure 1: **A** illustrates the growth in the number of floating-point operations (FLOPs) required for training deep models, starting from the inception of AlexNet (Krizhevsky et al., 2012). **B** depicts the corresponding increase in training costs (USD – 2020). **C** and **D** show the same trends but specifically for computer vision models. The data used in this analysis is sourced from Sevilla et al. (2022).

number of features describing both the operation and the operands, as well as the total number of available kernels. [2] While hand-written heuristic rules have shown some success in optimizing kernel selection, they are imperfect and their design requires significant time and effort from hardware experts and kernel writers. Moreover, such heuristics need to be manually adjusted or written anew for each new GPU architecture or firmware release. Given the rapid advancements in AI hardware and frequent firmware updates aimed at supporting the continuously evolving AI technologies, this approach introduces additional complications and delays, increasing the cost and complexity of the release process.

In this paper, we frame the kernel selection problem as a sequence classification task, where a sequence describing a mathematical operation and its operands needs to be mapped to the fastest kernel. This makes the problem suitable for attention-based sequence processing models, such as transformers (Vaswani et al., 2017). However, it is essential that any model for kernel selection be deployed on the CPU to avoid using GPU resources that are meant for user training and inference workloads. This constraint limits the effectiveness of transformer models, as their parallelism, especially in attention mechanisms, cannot be fully exploited on a CPU. As a result, the inference latency of a transformer-based model for kernel selection on a CPU can undermine the advantages gained from optimizing kernel selection. To this end, the contributions of our paper are as follows:

1. We present $\Gamma$-*block*, an alternative to transformer block with lesser operations. Our experiments on the IMDB and Reuters text classification tasks demonstrate that substituting the conventional transformer block with a $\Gamma$-block considerably reduces inference time while preserving comparable accuracy in smaller models. Additionally, we observe a substantial decrease in training time, which is particularly important given the frequent retraining required for kernel selection models in response to continuous firmware and hardware updates.

2. Using $\Gamma$-blocks, we propose a powerful yet lightweight model for kernel selection, named the *Reformer*. To train Reformer to predict the optimal kernel, we present a novel approach that assigns optimality probabilities based on kernel runtimes, as opposed to using one-hot probabilities that are highly sensitive to noise in runtime measurements.

3. We demonstrate Reformer's efficacy by deploying it to optimize convolution kernel selection in MIOpen (Khan et al., 2019), AMD's open-source library of high-performance compute kernels for machine learning operations. Our results on the AMD Instinct™ MI100

---

[2]For example, convolution operations are defined by features such as padding size, stride length, and dilation, and they involve tensors as operands, which are themselves characterized by multiple features, such as dimensionality, data types, and layouts. In addition, numerous algorithms exist for computing convolutions, each of which may be implemented using a range of optimized GPU kernels, whose performance depends heavily on the underlying hardware. Given all these variables, performing an exhaustive search over all possible convolution configurations to identify the optimal kernel for each problem configuration and hardware setup is impractical. And even if it were practical, the memory footprint of the mapping from convolution problems to best kernels would make this approach inefficient.

GPU show that the Reformer model outperforms MIOpen's hand-written heuristics, speeding up convolution operations by at least threefold. Furthermore, it streamlines and accelerates the release process for AI hardware and software, since training a neural network with automatic procedures like gradient descent is significantly less labor-intensive compared to manually writing and calibrating hand-tuned heuristics.

4. Deep learning's success is heavily owed to the efficiency enabled by GPUs. As the field advances, it's crucial to maintain this efficiency in GPU-accelerated AI computing. While significant contributions have been made by kernel designers towards this goal, our research showcases the alternative pathway: using deep learning methods to optimize deep learning hardware, which directly involves the deep learning community itself in GPU optimization.

This paper is structured as follows: Section 2 provides an overview of the kernel selection problem. Section 3 introduces the Reformer model, constructed using $\Gamma$-blocks, and demonstrates its performance on the IMDB and Reuters text classification datasets. In Section 4, we apply the Reformer model to optimize convolution kernel selection within MIOpen, presenting a novel method for estimating the likelihood of a kernel being optimal based on runtime data, followed by our results. Finally, Section 5 concludes with a brief review of related literature on kernel optimization, as well as works on transformers and ResNets, both of which share design similarities with our model.

## 2 BACKGROUND

Deep learning frameworks, such as PyTorch (Paszke et al., 2019) and Tensorflow (Abadi et al., 2016), represent neural networks as dataflow graphs, with each node corresponding to a specific mathematical operation, such as convolution, matmul, or softmax. These high-level operations are subsequently compiled into hardware-specific high-performance primitives, known as *kernels*, for GPU-accelerated computation. To achieve this, frameworks rely on kernel libraries, such as MIOpen (Khan et al., 2019), an open-source kernel library from AMD, and cuDNN (Chetlur et al., 2014), Nvidia's kernel library specific to Nvidia GPUs.

In this paper, we focus specifically on optimizing kernel selection for convolution operations, though the proposed model is general and can be deployed to optimize any operation. Both MIOpen and cuDNN offer a range of highly optimized kernels for convolution operations, which serve as the backbone of Convolutional Neural Networks (CNNs). CNNs have been at the forefront of many breakthroughs in computer vision (Lecun et al., 1998; Krizhevsky et al., 2012; Simonyan & Zisserman, 2014; Szegedy et al., 2015; He et al., 2016), primarily because convolution naturally encodes the spatial equivariance bias which facilitates the learning of general purpose visual representations Raghu et al. (2021). Therefore, despite the recent introduction of transformers (Vaswani et al., 2017) to the vision world (Dosovitskiy et al., 2020), CNNs continue to maintain a critical position in this domain, either on their own or as extensions to vision transformers (Dai et al., 2021; Xiao et al., 2021; Wu et al., 2021; Graham et al., 2021). Thus, improving the computational efficiency of CNNs can reduce the time and cost of training computer vision models while also enhancing energy efficiency and reducing the carbon footprint of data centers used for hosting, training, and serving deep learning models (Strubell et al., 2020; Patterson et al., 2022; Dodge et al., 2022). Beyond this, compute-efficient CNNs can also improve the inference speed and memory footprint of vision models deployed on edge devices, thus facilitating the adoption of deep neural networks in resource-constrained applications, such as robotics, smartphones, and IoT (Canziani et al., 2016).

Since convolutions are one of the primary operations in a CNN, the computational efficiency, evaluation time, and energy consumption of a CNN – both during training and inference – are heavily influenced by the kernels provided by the underlying kernel library for these operations. In particular, MIOpen employs hand-written rule-based heuristics specific to each GPU skew and firmware version to determine the most suitable kernel to serve from its pool.[3] However, these heuristics are imperfect, thus often leading to MIOpen serving suboptimal kernels to the upstream library, slowing down convolution operations. Since these suboptimal kernels may be invoked tens of thousands of

---

[3]MIOpen also maintains a small internal lookup table, referred to as `FindDB`, which catalogs "common" convolution problems and their corresponding fastest kernels, derived through exaustive benchmarking (Khan et al., 2019). For a given convolution problem, MIOpen first checks `FindDB`; if a match is found, it retrieves the kernel from the table. Otherwise, MIOpen uses heuristics to select a kernel.

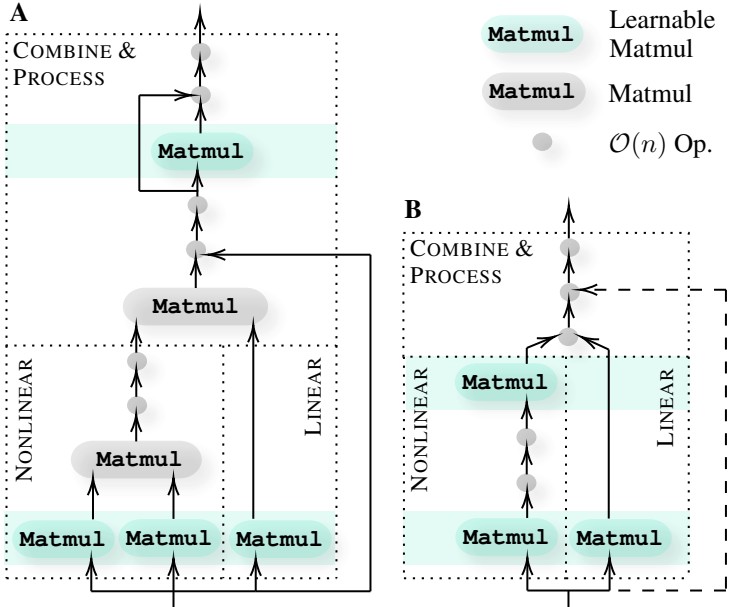

Figure 2: Wireframe diagrams of the transformer block (**A**) from Vaswani et al. (2017) and the Γ-block (**B**) proposed in this paper. Both blocks process input through nonlinear and linear pathways, subsequently combining and further processing the outputs of these pathways. The Γ-block is designed to have fewer operations and a narrower structure compared to the transformer block. Specifically, the Γ-block includes half as many matrix multiplications (matmuls) as the transformer block (3 versus 6), while maintaining the same depth in terms of learnable layers (2 versus 2). Additionally, the transformer block features six elementwise ($\mathcal{O}(n)$) operations (two nonlinearities, two norms, and two additions), whereas the Γ-block includes only three to four such operations (one nonlinearity, one norm, and one to two additions, depending on the presence of a skip connection).

times during training and potentially millions of times when the model is served, this accumulated latency can cause significant slowdowns, increased costs, and higher energy consumption. Moreover, maintaining hand-tuned heuristics is both time-consuming and error-prone, as these heuristics must be manually designed and adjusted by experts whenever new hardware or firmware updates are released. In this paper, we present the *Reformer* neural network, which we show can address all of these concerns: it is not only significantly more accurate than hand-tuned heuristics but can also readily be adapted across different hardware architectures and firmware releases.

## 3  REFORMER MODEL

We define a reformer model as a composition of Γ-*blocks*. That is, a reformer model $\Gamma : \mathbb{R}^d \to \mathbb{R}^C$ of depth $L$ is a transformation defined as[4]

$$\Gamma(x) = \Xi \left( \Gamma_L \left( \dots \Gamma_2 \left( \Gamma_1 (\Upsilon(x)) \right) \dots \right) \right),$$

where $\Gamma_\ell : \mathbb{R}^{d_{\ell-1}} \to \mathbb{R}^{d_\ell}$, for $\ell = 1, \dots, L$, denotes a Γ-block, $\Upsilon : \mathbb{R}^d \to \mathbb{R}^{d_0}$ represents an initial transformation on the input $x$, and $\Xi : \mathbb{R}^{d_L} \to \mathbb{R}^C$ denotes a transformation on the output of the final Γ-block, $\Gamma_L$. Denoting $\mathbb{R}^{d_0} \ni x_0 = \Upsilon(x)$, we define the Γ-blocks $\Gamma_\ell$ as:

$$\Gamma_\ell(x_{\ell-1}) = (A_1)_\ell \, x_{\ell-1} + (b_1)_\ell + (A_3)_\ell \, \mathcal{T}_\ell(x_{\ell-1}) + (b_3)_\ell := x_\ell. \tag{1}$$

Here $(A_1)_\ell \in \mathbb{R}^{d_{\ell-1} \times d_\ell}, (b_1)_\ell \in \mathbb{R}^{d_\ell}, (A_3)_\ell \in \mathbb{R}^{d'_\ell \times d_\ell}$, and $(b_3)_\ell \in \mathbb{R}^{d_\ell}$ are learnable parameters of the layer, and $\mathcal{T}_\ell : \mathbb{R}^{d_{\ell-1}} \to \mathbb{R}^{d'_\ell}$ denotes a fully-connected network, consisting of ReLU, normalization (batchnorm or layernorm) and dropout. Figure 2 shows a comparison of the transformer

---

[4]In the context of the kernel selection problem, $d$ corresponds to the number of features that characterize an operation and its operands, while $C$ represents the number of kernels available for that operation in the library.

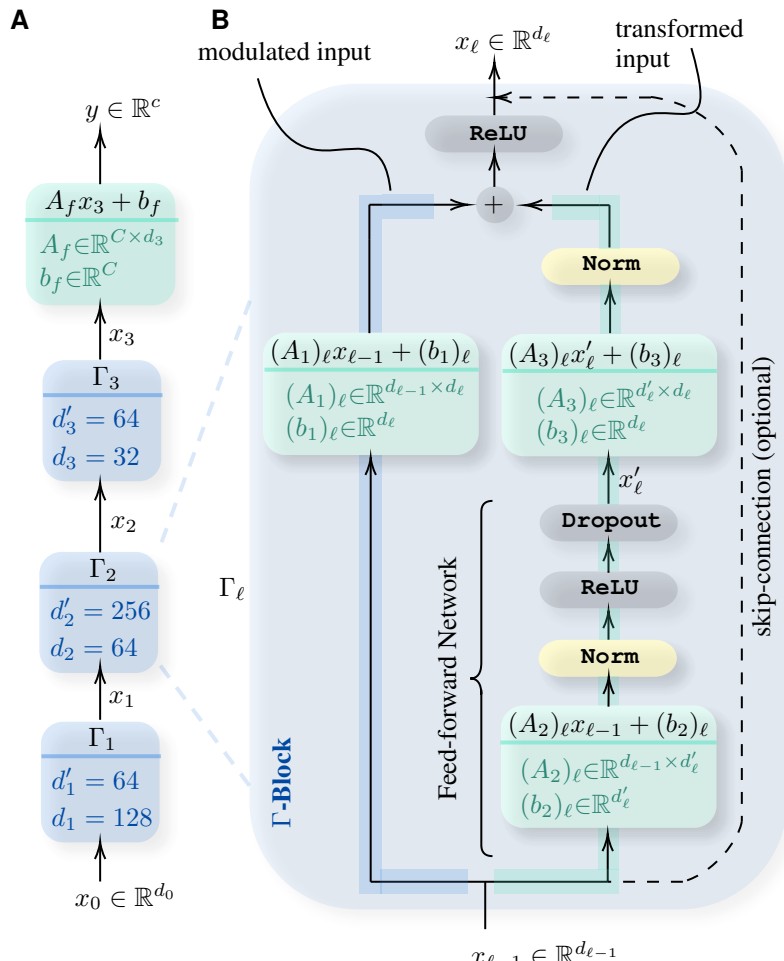

Figure 3: **A** shows the architecture of the proposed Reformer model for convolution kernel selection in MIOpen. At the heart of the network are three Γ-blocks – $\Gamma_1, \Gamma_2$ and $\Gamma_3$ – where the size of each block $\Gamma_\ell$ is controlled by two attributes: $d'_\ell$ and $d_\ell$. Each Γ-block $\Gamma_\ell$ features two pathways that independently process the input to the layer, $x_{\ell-1}$, as shown in **B**. The pathway to the right (shaded cyan) processes the input through a feedforward network followed by an affine mapping, and the other pathway (shaded blue) allows the input signal to flow unhindered except for an affine projection. The outputs of these pathways are then combined element-wise and passed through a ReLU activation function. Although the Γ-block architecture resembles blocks used in ResNet variants, its original inspiration was the transformer block, with the goal of simplifying it for CPU compute.

block from Vaswani et al. (2017) to the Γ-block. It can be seen that the Γ-block is narrower, with half the number of matmul operations as the transformer block, while being equally deep in terms of the number of learnable layers. Moreover, the Γ-block entails fewer element-wise operations. However, at a higher level, one can see that both the Γ-block and the transformer block share a similar structure: they each generate independent nonlinear and linear representations of the input, combine these representations, and then process the combined output. This process is detailed in Figure 3(**B**) for the Γ-block. The cyan-shaded pathway projects the input $x_{\ell-1} \in \mathbb{R}^{d_{\ell-1}}$ onto a nonlinear manifold in $\mathbb{R}^{d'_\ell}$ defined by $\mathcal{T}_\ell$, which is described in $\mathbb{R}^{d_\ell}$ through an affine map defined by $(A_3)_\ell$, and $(b_3)_\ell$. The blue-shaded pathway, on the other hand, uses an affine map defined by $(A_1)_\ell$, and $(b_1)_\ell$ to directly describe the input in $\mathbb{R}^{d_\ell}$, without utilizing an intermediate nonlinear representation. These two representations are additively combined, and subsequently ReLU-ed.

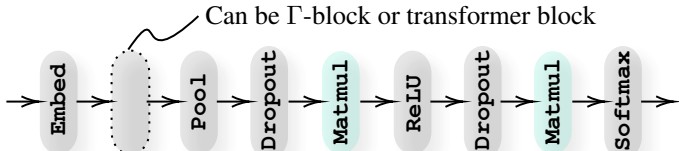

Figure 4: A small text classification model inspired from the transformer architecture. The unnamed block can be either the transformer block or Γ-block. The model's performance on the IMDB and Reuters datasets in both scenarios is provided in Table 1.

Table 1: Comparison of the Γ-block and transformer block performances on the IMDB and Reuters text classification datasets. The full model architecture is illustrated in Figure 4. All tests were conducted on an Intel®Core™ i7-6700K CPU.

|  |  | IMDB | Reuters |
|---|---|---|---|
| **Γ-Block (Ours)** | Train Accuracy | 99.99% | 94.32% |
|  | Test Accuracy | 87.07% | 75.96% |
|  | Training Time | 9.28 sec/epoch | 2.99 sec/epoch |
|  | Inference Time | 12.04 sec | 12.27 sec |
| **Transformer Block** | Train Accuracy | 100% | 96.04% |
|  | Test Accuracy | 87.14% | 76.24% |
|  | Training Time | 23.05 sec/epoch | 7.31 sec/epoch |
|  | Inference Time | 15.19 sec | 15.32 sec |

## 3.1 CASE STUDY: TEXT CLASSIFICATION

We present a case study on the IMDB and Reuters text classification datasets to demonstrate the Γ-block's performance and compare it to that of the transformer block. To ensure a level playing field for a direct model comparison, we design a small model inspired by the transformer model, as shown in Figure 4, that can be instantiated with either a transformer block or Γ-block. Our findings, summarized in Table 1, show that the model utilizing the Γ-block achieves comparable accuracy to its transformer block counterpart while being significantly faster to train and having a considerably lower inference time.

It is important to note that these experiments were conducted using a small model. While results may vary for larger model sizes, we intentionally chose a smaller size to align with the scale suitable for deployment in a kernel selection application. Kernel selection takes place at the lowest level of the deep learning stack, where minimizing resource consumption is essential in order to maximize the amount of resources available to upstream libraries.

Table 2: Example of a convolution problem with six applicable kernels, labeled A through F for readability. Kernel E is identified as optimal, with a runtime of $0.0036$ ms. The table also illustrates various methods for assigning probabilities to these kernels, indicating their likelihood of being optimal. This demonstrates the various ways that kernel selection can be formulated as a classification problem aimed at predicting the label of the optimal kernel.

|  | Kernel Runtime | One-hot Probability | Softmax Probability | Ratio-preserving Probability (ours) |
|---|---|---|---|---|
| Kernel A | 0.0136 ms | 0 | 0.19048 | 0.17384 |
| Kernel B | 0.0507 ms | 0 | 0.18354 | 0.04645 |
| Kernel C | 0.0282 ms | 0 | 0.18772 | 0.08358 |
| Kernel D | 0.0573 ms | 0 | 0.18233 | 0.04112 |
| Kernel E | 0.0036 ms | 1 | 0.19239 | 0.65290 |
| Kernel F | 1.1114 ms | 0 | 0.06354 | 0.00212 |

# 4 CONVOLUTION KERNEL SELECTION

We show our particular instantiation of the Re-former architecture for convolution kernel selection in Figure 3(**A**). We use three $\Gamma$-blocks, i.e., $L = 3$, and we choose $\Upsilon$ to be z-score normalization and $\Xi$ to be a learnable linear layer defined as $\Xi(x_L) = \sigma\left(A_f(x_L) + b_f\right)$, where $\sigma$ is the softmax function. These simple choices for $\Upsilon$ and $\Xi$ were made to keep the model's compute footprint low. Although more complex layers, such as an input embedding for $\Upsilon$, yielded marginal improvements in accuracy, they came at the cost of increased inference time, ultimately negating the benefits of accuracy improvement. We interpret the model's output, $\mathbb{R}^C \ni y := \Gamma(x)$, as a "probability distribution" over the kernels, where $y_i$ indicates the likelihood that the $i$-th kernel is optimal for solving the convolution problem with features $x$.

We treat kernel selection as a classification task, where the objective is to identify the label of the optimal kernel from a set of candidates. Typically, classification models are trained to generate a one-hot distribution, concentrating all probability mass on the correct class (which in this case would be the label of the optimal kernel) and treating all other classes as equally undesirable. However, this modeling assumption oversimplifies the problem of kernel selection. For instance, consider a convolution problem that can be solved by six different kernels, with runtimes listed in Table 2. Kernel E, with the shortest runtime, is clearly the optimal choice. However, not all other kernels are equally undesirable; some are significantly less desirable than others. For instance, Kernel F is considerably slower than Kernel A and is thus much less preferable. This distinction becomes crucial in cases where two kernels have nearly identical performance on a set of problems, and random noise causes one kernel to appear optimal in some of the problems and the other to appear optimal in others. Assigning one-hot probabilities can mask the fact that the two kernels are actually close in performance and either may be regarded as optimal.

One can find a better prior, capturing kernel rankings as a continuous function of their runtimes, by applying softmax on the runtimes, as shown in Table 2. However, this yields probabilities related to the kernel runtimes on a logarithmic scale (because of the exponentiation operation involved in softmax), which can often be undesirable, giving two kernels with significantly different performance similar probability weights. For example, in Table 2, Kernel E and Kernel D have very similar softmax-assigned probabilities even though Kernel E is about 16 times faster than Kernel D.

To address this, we propose a novel approach to convert kernel runtimes into probabilities that follow the ratios between kernel runtimes. Given a vector $t$ of all kernel runtimes for a problem, we compute

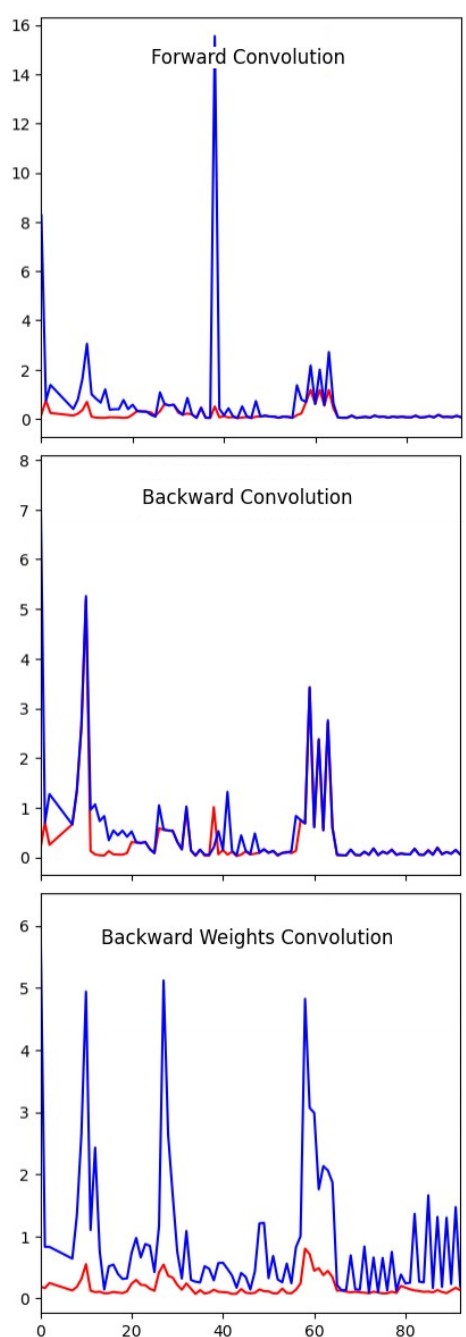

Figure 5: Performance comparison between the Reformer model (red) and hand-written rules-based heuristics (blue) for convolution kernel selection on the AMD Instinct™ MI100 GPU. The x-axis represents unique convolution problems, and the y-axis shows the runtime (in microseconds) for each kernel.

probabilities $p$ as follows:

$$p_i = \frac{1}{\sum_{k=1}^{C} \max(t)/t_k} \left( \frac{\max(t)}{t_i} \right).^5$$

Here, $C$ is the number of kernels available in the library, where we take $t_i = \infty$ if the $i$-th kernel is not applicable to the problem. This ensures that $p_j/p_i = t_i/t_j$, meaning if the $i$-th kernel is $\beta$ times faster than the $j$-th kernel, then $p_i$ will be $\beta$ times lower than $p_j$. For example, using this method, Kernel D is assigned a probability mass about 16 times lower than Kernel E, inline with their runtimes.

We use the probabilities $p$ as the ground-truth to train our Reformer model for kernel selection. Specifically, we define the loss function as the cross-entropy between $p$ and the predicted distribution $y$ output by the Reformer model:

$$\mathcal{L}(x) = H(p, y) = -\sum_{i=1}^{C} p_i \log y_i.$$

To generate the training/testing dataset, we used a collection of 500,000 convolution problems curated internally at AMD for the purposes of tuning and QA testing. For each of these convolution problems, we evaluated every applicable kernel inside MIOpen on the AMD Instinct™ MI100 GPU, and created a dataset mapping convolution problems to optimal kernels. We subsequently split this dataset into an 80% training set and a 20% test set.

We note that our data collection process encountered some noise due to distributed benchmarking across machines in different locations. Despite ensuring that all machines had identical hardware and software environments, perfect consistency could not be guaranteed. Moreover, minor variations in external factors, such as ambient temperature around the machines, along with other sources of random machine noise, also introduced some inconsistencies into the dataset. Nonetheless, we trained the Reformer model on this dataset, achieving an overall test accuracy of 91.7% in predicting the optimal kernel. For comparison, we also trained an off-the-shelf ResNet18 model on the same dataset. Despite being much larger in size, it achieved only a marginally higher test accuracy of 92.1%.

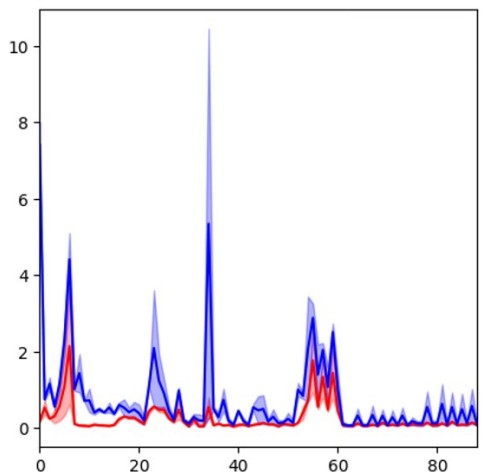

Figure 6: Comparison of the Reformer model (red) and MIOpen's hand-tuned heuristics (blue) for kernel selection across 100 different convolution problems, averaged over FWD, BWD, and WRW directions.

Figure 5 illustrates the Reformer model's performance on 100 different convolution problems in all three directions: Forward (FWD), Backward (BWD), and Backward with Weights (WRW). These problems were proposed by an engineer with no prior knowledge or involvement in this study, making them effectively out-of-distribution for the model. Despite this, the Reformer model's predicted kernels demonstrated significant efficiency gains over the heuristics in MIOpen. Specifically, the Reformer model was 3.40 times faster in the FWD direction, 1.51 times faster in the BWD direction, and 5.46 times faster in the WRW direction compared to the heuristics in MIOpen on the AMD Instinct™ MI100 GPU. Overall, as shown in Figure 6, integrating the Reformer model for convolution kernel selection into MIOpen led to an average speedup of approximately 3x across all convolution directions.

This particular instantiation of the Reformer model contains 80,978 parameters, requiring only about $160\,\text{kB}$ of memory when stored in FP16 precision, with potential further reductions through quantization. Its inference time on the CPU clocks at about $49\,\mu\text{s}$, which is typically much shorter than the runtime of the slowest kernel for most convolution problems. Importantly, this inference is only

---

[5]It is straightforward to show that the expression for $p_i$ adheres to the Kolmogorov Axioms for $t_i > 0$.

needed during the initial pass of a deep learning model to identify the optimal kernels. Once identified, these kernels are cached and the Reformer model is unloaded from memory. For all subsequent passes – which can range from tens of thousands during training to millions when a deep learning model is deployed – the kernels are simply retrieved from the cache. Consequently, the inference time of the Reformer model becomes essentially negligible when amortized over thousands, if not millions, of runs of a deep learning model. Moreover, the memory footprint is also significantly reduced after the first run, as the model is unloaded from memory and replaced by a cache/dictionary.

It is worth reiterating here that the Reformer model is not just considerably more accurate but also more scalable than rules-based heuristics. In contrast to hand-written rules, which may require extensive revisions for new firmware releases and complete rewrites by experts for new GPU releases, the Reformer model can be easily deployed and updated using automated routines. It simply takes benchmarking a set of convolution problems on the new hardware and/or firmware[6] and training the model on the resulting dataset using the gradient descent algorithm. Because of this automation, the Reformer-based kernel selection is also less error-prone and more consistent, as illustrated by Figure 6, which shows significantly more performance spikes and variation when kernels are chosen using rule-based heuristics despite the considerable effort put into their design by kernel developers and QA teams.

## 5 RELEVANT WORK

### 5.1 KERNEL OPTIMIZATION

Previous work on optimizing the performance of compute kernels has mostly concerned itself with methods for tuning/optimizing kernel parameters (Guerreiro et al., 2015; Lloyd et al., 2018; Gale et al., 2020; Bhaskaracharya et al., 2020). Additionally, the problem of kernel scheduling has also received some attention. For instance, Shekofteh et al. (2019) propose a method to select kernels based on whether they are compute-bound or memory-bound in order to optimize concurrent kernel scheduling on modern GPUs. Ahmed et al. (2022) propose a machine learning approach for selecting the best kernels to fuse together so as to maximize GPU utilization. The work of Jeon et al. (2022) presents a system for selecting the optimal backend to pick a kernel for deep learning operations. Some other relevant works include: Oyama et al. (2018), Liu et al. (2021a), Xiao et al. (2020), and the review article on parallel deep learning by Ben-Nun & Hoefler (2019). However, while these works share overlapping themes with our research, to the best of our knowledge, none of them specifically address the problem of mapping high-level deep learning operations to fastest GPU kernels from a pool of kernels, particularly using deep learning methods.

### 5.2 RESIDUAL NETWORKS

Deep Residual Networks (ResNets), introduced by He et al. (2016), pioneered the use of *skip connections*, which have since become a common fixture in neural network architectures. Despite its design inspiration from Transformers, the Reformer model also shares some architectural similarities with ResNets, including skip connections. However, unlike ResNets, the Reformer features skip connections only between blocks rather than within them. Additionally, whereas ResNets use convolutional layers within their blocks, the Reformer model employs fully connected layers similar to those in attention modules of Transformers.

### 5.3 TRANSFORMERS

Introduced as a machine translation model by Vaswani et al. (2017), the transformer architecture has since seen wide success in other applications of deep learning as well (Devlin et al., 2018; Carion et al., 2020; Dosovitskiy et al., 2020). However, the necessity of the core architectural element of the transformer model – the powerful yet compute-expensive self-attention mechanism – has since been brought to question. Recent works have replaced the attention blocks with units composed of

---

[6]Kernel benchmarking for data collection can be streamlined using a tool like MITuna, an open-source library developed at AMD that enables kernel benchmarking to be configured and monitored through user-friendly Jenkins pipelines, with the ability to distribute tasks via SLURM across multiple machines that meet hardware requirements, while automatically setting up the necessary software environment.

MLPs, and achieved comparable performance (Liu et al., 2021b; Tolstikhin et al., 2021). Our work also aligns with these works, but our architecture is optimized specifically for CPU deployment. Furthermore, while many of these existing architectures were developed for language modeling tasks, our architecture is designed specifically for the task of kernel selection.

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
