# OpenReview forum: "Reformer: A Deep Learning Model for Runtime Selection of Convolution Kernels"
_ICLR.cc/2025/Conference — Submitted to ICLR 2025_

### Official Review · Reviewer_S3ew · 2024-10-21

**Soundness:** 1
**Presentation:** 2
**Contribution:** 1
**Rating:** 3
**Confidence:** 4

**Summary:**

A sequence of kernel programs are executed to run a model on processors. To maximize the model execution efficiency, usually there will be many candidates for each kernel and the best one will be picked up during runtime, and this process is called kernel selection. This paper proposes a new transformer-variant called Reformer to model the kernel selection as a sequence classification problem and run the proposed model on CPU to save the GPU resources. Instead of using one-hot label or softmax sore as the optimization target, authors proposed to use optimality probabilities. Experiments show that the proposed kernel selection model selects better kernels than the MIOpen hand-written heristics for the convolution kernels on AMD MI100 GPU.

**Strengths:**

1. The paper is easy to understand.
2. The proposed method can be executed only on CPU, which saves the GPU resources during kernel selection.

**Weaknesses:**

1. The motivation to use a new transformer-variant instead of original transformer architecture is weak.
2. The evaluation is weak as well.

See comments below for details.

**Questions:**

1. The motivation to use a new transformer-variant instead of original transformer architecture is weak.
From Table 1, we can see that the proposed block is about 25% faster than transformer block. In Section 4, the authors report that the proposed kernel selection model using the new block design only takes 49 macro-seconds. If we directly use transformer model with similar number of parameters as the proposed model, can we achieve similar selecting performance (like <0.1 ms)? If so, why do we need to use a new architecture design when the model execution efficiency is not a bottleneck?

2. The evaluation is weak as well.
The paper only evaluates the hand-tuned heristics baselines. But the kernel selection (or operator/kernel tuning) problem has been studid widely in deep learning compiler domain. Many other baselines are used like using simple linear layers to predict latency, use xgboost to predict or select kernels, or use evolutionary algorithms to select best kernels (see AutoTVM [1]). The paper does not compare with these baselines.
If the main novelty of the paper is to propose a new transformer-variant, then a bunch of other transformer-variants should be compared as well.

[1] TVM: An Automated End-to-End Optimizing Compiler for Deep Learning

---

### Official Review · Reviewer_S1hz · 2024-10-25

**Soundness:** 2
**Presentation:** 2
**Contribution:** 2
**Rating:** 3
**Confidence:** 3

**Summary:**

The Reformer paper introduces a deep learning model designed to optimize convolution kernel selection for GPUs, significantly improving runtime efficiency. By utilizing the novel Γ-block, which reduces the number of matrix operations compared to traditional transformer blocks, Reformer accelerates kernel selection while maintaining comparable accuracy. Integrated into AMD's MIOpen, the model achieves a 3x speed-up in convolution operations. It replaces labor-intensive manual heuristics with a machine learning approach, treating kernel selection as a classification task using runtime data to predict optimal kernels. The paper also highlights how Reformer selects convolution kernels to enhance runtime performance.

**Strengths:**

The paper presents the following strengths. Its introduction of the Γ-block reduces the number of matrix operations compared to traditional transformer blocks, leading to significant improvements in runtime efficiency, including a 3x speed-up in convolution kernel selection. The novel approach frames kernel selection as a machine learning classification task, replacing labor-intensive hand-tuned heuristics with a more automated and accurate method. The model is also lightweight and optimized for CPUs, which frees up GPU resources for training and inference, making it ideal for practical deployment. Its adaptability to various hardware architectures, with the ability to automatically retrain when firmware or hardware changes, adds to its scalability across different systems.

**Weaknesses:**

The paper has weaknesses that limit its broader impact. Its scope is primarily focused on optimizing convolution kernel selection, and it does not explore the generalizability of the approach to other deep learning operations or tasks. Additionally, while the Γ-block demonstrates efficiency gains, the paper lacks a detailed comparison of computational complexity with other cutting-edge methods, like Transformers, making it harder to evaluate its scalability on larger models and datasets. Testing is also primarily limited to GPUs, with little discussion of how the model performs on other hardware, such as GPUs or TPUs. Furthermore, while the model streamlines kernel selection, the initial benchmarking and training process needed to build the dataset may introduce overhead when adapting to new environments. The approach depends on accurate runtime data, which can be noisy or inconsistent, potentially affecting the robustness of the model in real-world scenarios. The approach also doesnt compare with other similar work such as AutoTVM.

**Questions:**

A few comments and questions to improve the paper:
In Figure 3, the way of depicting a linear layer or matrix multiplication could be simplified for clarity, reduce amount of text in the diagram.
Additionally, it's worth noting that there is already a transformer variant named "Reformer," so the author maight want to rename it.
How does the Reformer model compare with other compilation techniques, such as Auto-TVM or the PyTorch compiler?
Why does the paper focus solely on optimizing CNN kernels, rather than exploring optimizations for other types of deep learning operations?

---

> ### Author Response · Authors · 2024-12-03
> **Response to Reviewer S1hz**
>
> Thank you for the time and effort you invested in reviewing our submission. Unfortunately, due to extensive commitments that required our immediate attention during the review discussion period, we were unable to provide timely responses to your comments or address them in our work in time for this submission.
>
> Our focus is on convolution kernels to demonstrate the approach; however, the method is broadly applicable to other operations. While adding results for additional operations would be a valuable addition, it is also a time-intensive task that we may undertake in a future revision. Regarding results on other hardware, we acknowledge this as a consistently noted shortcoming and plan to address it by including comparisons on additional hardware in a future revision.
>
> We remain committed to improving our work and addressing your valuable feedback in subsequent revisions. Thank you again for your thoughtful comments and for helping us enhance the quality of our research.

---

### Official Review · Reviewer_rU2B · 2024-11-01

**Soundness:** 3
**Presentation:** 3
**Contribution:** 3
**Rating:** 5
**Confidence:** 3

**Summary:**

Deep learning models rely heavily on efficient convolution operations, and choosing the right GPU kernel for these operations is crucial for performance.  Traditionally, developers have relied on hand-crafted rules to select kernels, a process that's time-consuming, prone to errors, and often tied to specific hardware.  This paper introduces "Reformer," a model that intelligently automates this kernel selection process.  Instead of relying on fixed rules, Reformer learns to predict the optimal kernel by treating the problem as a sequence classification task.  Cleverly, it performs this analysis on the CPU, freeing up valuable GPU resources for the actual deep learning computations.

Reformer presents a few key innovations: a streamlined building block called the "Γ-block" that's efficient enough for CPU inference; a unique training method called "ratio-preserving probability training" which produces a more robust kernel selection model; and seamless integration with AMD's MIOpen library.  In tests on an AMD MI100 GPU, Reformer demonstrated a significant speed boost—around 3x faster—compared to existing hand-tuned methods.  This suggests that Reformer offers a promising path towards more adaptable and performant deep learning systems.

**Strengths:**

**Originality:** The core idea of using a deep learning model for kernel selection is novel. While previous work has explored machine learning for kernel scheduling or parameter tuning, this paper tackles the more challenging problem of dynamically selecting the best kernel from a pool of options based on the specific operation and its operands.  The introduction of the Γ-block, specifically designed for CPU inference in this context, also contributes to the originality.

**Quality:** The paper demonstrates a thorough understanding of the kernel selection problem and the limitations of existing methods. The proposed Reformer model is well-motivated and carefully designed. The experimental results, including the integration with MIOpen and the 3x speedup achieved, provide strong evidence of the effectiveness of the approach.  The comparison with a ResNet18 baseline further strengthens the argument for Reformer's efficiency.

**Clarity:** The paper is generally well-written and easy to follow. The figures effectively illustrate the architecture of the Reformer model and the Γ-block, and the tables clearly present the experimental results. The paper provides sufficient background information on kernel selection and related work to contextualize the contributions.

**Significance:** The work addresses a critical bottleneck in GPU-accelerated deep learning: efficient kernel selection. The proposed method has the potential to significantly improve the performance and reduce the cost of deep learning training and inference. The automation offered by Reformer also simplifies the development and deployment process for GPU software and hardware.  Furthermore, the paper's approach of using deep learning to optimize deep learning hardware opens up exciting new possibilities for future research in this area.

**Weaknesses:**

**Hardware Specificity:**  The experimental evaluation primarily focuses on convolution kernels within MIOpen and AMD GPU. While this is a significant proof, it would be beneficial to assess Reformer's generalizability by applying it potentially to other kernel libraries like cuDNN.
Or please add discussion about how the proposed approach can be applied to other libraries if there is any difficulty in testing on Nvidia GPU. It's important to investigate how Reformer performs on different GPU architectures (both AMD and Nvidia) and across varying hardware generations. This would demonstrate the broader applicability of the proposed method.

**Ablation Study:**  A more detailed ablation study would be helpful in understanding the contribution of each component of the Reformer model.  For example, how does the performance change with varying numbers of Γ-blocks?  Is the ratio-preserving probability training truly superior to softmax in all cases, or are there scenarios where softmax performs comparably?  From table 2, the one-hot, softmax and ratio-preserving probability actually concur with each other? (ratio-preserving probability indeed has more salient probability value)

**Scalability:** The paper focuses on a small model size, justifiable for kernel selection. However, it's crucial to discuss the scalability of the approach. How does the training time and inference latency of Reformer scale with increasing model complexity (i.e., for larger numbers of kernels and input features)? This information would be relevant for considering the potential application of Reformer to other domains beyond kernel selection.

**Noise Robustness:**  While the ratio-preserving training aims to address noise in runtime measurements, a more direct analysis of Reformer's robustness to noise would be valuable.  For instance, artificially introducing varying levels of noise into the training data and measuring the impact on accuracy would provide a clearer picture of the method's resilience to real-world variations in runtime measurements.

**Questions:**

- Questions listed in weakness section
- You are talking about convolutional kernel selection, I feel it a bit strange to use text classification as case study, wouldn't the image classification problem be more natural? This could also expand the generalizability of the approach. a CIFAR classification would be sufficient

---

> ### Author Response · Authors · 2024-12-03
> **Response to Reviewer rU2B**
>
> Thank you for the time and effort you dedicated to reviewing our submission. Unfortunately, due to extensive commitments that required our immediate attention during the review discussion period, we were unable to provide timely responses to your comments or incorporate them into our work in time for this submission.
>
> Regarding your questions:
>
> **Performance results from other GPUs:** We agree with your comment and will include performance results from additional GPUs in a future revision. This is a point consistently raised by other reviewers as well, and we are committed to addressing it comprehensively in our future work.
> **Ablation study:** A detailed ablation study would indeed be an interesting addition. Generally, performance increases as the number of blocks is increased, up until the model begins to overfit. However, inference time also scales with the number of blocks, which is critical to keep low for kernel selection tasks. As a result, the number of blocks cannot be increased arbitrarily.
> Regarding the ratio-preserving probability, it has shown superior performance compared to softmax on the dataset used in the paper. We plan to produce results for additional datasets in a future revision.
> On your third question, "From Table 2, the one-hot, softmax, and ratio-preserving probability actually concur with each other?": I am unclear about the exact nature of your concern here. Could you please clarify?
> **Focus on kernel selection:** The primary focus of this paper is kernel selection, particularly for convolution operations. While results from other tasks could be useful, they would extend beyond the intended scope of this work.
> **Artificially adding noise:** This is an excellent suggestion, and we appreciate the idea of artificially introducing varying levels of noise to compare the softmax and ratio-preserving methods. We will consider incorporating this analysis in a future revision.
>
> We are committed to improving our work and addressing your valuable feedback in subsequent revisions. Thank you again for your thoughtful comments and for helping us improve the quality of our research.

---

> > ### Author Response · Authors · 2024-12-03
> > **Answer to last question**
> >
> > Regarding your last question, *"You are talking about convolutional kernel selection. I feel it a bit strange to use text classification as a case study. Wouldn't the image classification problem be more natural?"*
> > Since the task is to select optimal kernels for operations, text classification aligns closely with the nature of the problem and serves as a more relevant case study. Language is a 1D sequence, somewhat similar to the sequence of features used to describe an operation.

---

### Official Review · Reviewer_4ec1 · 2024-11-02

**Soundness:** 3
**Presentation:** 3
**Contribution:** 3
**Rating:** 6
**Confidence:** 2

**Summary:**

In this paper, this paper propose a new deep learning model, "Reformer," for GPU kernel selection optimization.
Based on Γ-block, transformer reconstructs the kernel selection problem as a sequence classification problem, and works efficiently in CPU to provide faster inference time.
Experiments show that transformer performs on average three times faster than MIOPen's manual heuristic, and automates the kernel selection process to simplify the deployment of AI hardware and software.

**Strengths:**

1. The research is CPU-friendly design. Therefore, the research may efficiently use CPU resources.
2. The Reformer model reconstructs the kernel selection problem specifically for convolution operations as a sequence classification problem, and predicts the most suitable convolution kernel based on the features of the operation. This automates the selection process and quickly finds the optimal convolution kernel, improving overall model training and inference speed.

**Weaknesses:**

1. This study conducted experiments using a limited dataset and a restricted set of model comparisons. Are there no experiments conducted on other datasets?
2. Although a comparison with MIOpen's manual heuristics was performed, there is a lack of comparison with other state-of-the-art kernel optimization techniques, making it necessary to further substantiate Reformer’s performance improvements.

**Questions:**

1. It was previously mentioned that the structure of Reformer could be applied to other types of optimization problems. However, is there more concrete evidence that it guarantees performance improvements in kernel optimization for other operations as well?
2. Could you provide experimental results on other recent datasets or state-of-the-art models?

---

> ### Author Response · Authors · 2024-12-03
> **Response to Reviewer 4ec1**
>
> Thank you for the time and effort you invested in reviewing our submission. Unfortunately, due to extensive commitments that required our immediate attention during the review discussion period, we were unable to provide timely responses to your comments or address them in our work in time for this submission.
>
> 1. We haven’t run tests on other operations yet, but it would certainly be interesting to explore and obtain performance numbers. The methodology remains consistent—what holds for convolution also applies to other operations, such as GEMM. The features of a GEMM can be transcribed into a sequence in the same way as those of a convolution problem, and ratio-preserving probabilities can similarly be assigned to GEMM problems.
> 2. We agree with your comment and will include comparisons for datasets based on Navi and MI300 GPUs in a future revision.
>
> We remain committed to refining our work and incorporating your valuable feedback in a future revision. Thank you again for your thoughtful comments and for helping us improve the quality of our research.

---

### Official Review · Reviewer_rWk5 · 2024-11-02

**Soundness:** 3
**Presentation:** 3
**Contribution:** 3
**Rating:** 6
**Confidence:** 3

**Summary:**

This paper addresses the problem of optimizing GPU kernel selection for neural network operations, with a focus on convolution kernels in AMD's MIOpen library. The authors propose framing kernel selection as a sequence classification problem and introduce an efficient alternative to transformer block for CPU deployment. The proposed alternative, Γ-block, requires 3 matmuls instead of transformer's 6, making it more suitable for CPU inference where parallelism is limited. They build a lightweight model called Reformer by stacking multiple Γ-blocks and train it to predict the optimal kernels based on convolution problem features. To handle the challenge of noisy runtime measurements and the sensitivity of one-hot encoding, they introduce a novel probability assignment method that reflects the relative performance of kernels. Compared to existing hand-tuned heuristics, the Reformer model gives an average 3x speedup in conv ops on the AMD MI100 GPU.

**Strengths:**

* Optimizing GPU kernel selection is an important problem with practical implications for reducing compute and energy consumption of deep learning workloads.
* The paper presents a novel approach to kernel selection by recasting it as a sequence classification problem and introducing the Γ-block as a more CPU-efficient alternative to transformer blocks.
* Thorough experimental evaluation of Γ-block. The authors demonstrate comparable accuracy with reduced inference and training times on standard text classification datasets. Integrating Reformer to real-world convolutional kernel selection in MIOpen also shows significant performance improvements over existing heuristics.
* The paper is well-written and organized. The architectures of the Γ-block and Reformer model are clearly described with helpful figures.

**Weaknesses:**

* While the authors show significant improvements in convolutional kernel selection, the evaluation is limited to the AMD MI100 GPU and the MIOpen library. It is unclear how well the Reformer model generalizes to other GPU architectures or kernel libraries (e.g. NVIDIA). Broader evaluation across different hardware and software environments would strengthen the generalizability claim.
* The paper compares the Reformer model with hand-tuned heuristics and an off-the-shelf ResNet18 model but doesn't compare against other approaches like gradient boosting commonly used for kernel selection. Including such comparisons would provide a comprehensive assessment of Reformer's effectiveness.

**Questions:**

1. Have you tested the Reformer model and the Γ-block on other GPU architectures or with other kernel libraries? How does the model performance vary across different hardware platforms?
2. You mention that noise due to distributed benchmarking and environmental factors introduced inconsistencies in the dataset. How does this noise affect the model's predictions in practice, and what strategies could mitigate its impact?
3. Have you considered comparing the Reformer model with other models (e.g., XGBoost) for kernel selection? If so, how do they perform in terms of accuracy and inference time?

---

> ### Author Response · Authors · 2024-12-03
> **Response to Reviewer 4ec1**
>
> Thank you for the time and effort you invested in reviewing our submission. Unfortunately, due to extensive commitments that required our immediate attention during the review discussion period, we were unable to provide timely responses to your comments or address them in our work in time for this submission.
>
> Regarding your questions:
> 1.  The reformer model does generalize to other GPUs. For instance, we have successfully deployed it on the MI200 and MI300 GPUs. We will include results from this in a future revision.
> 2. In practice, the effect of noise is minimal when using ratio-preserving probability, and it can be further mitigated by utilizing larger datasets.
> 3. A comparison with classical techniques would be an interesting addition to the paper. We are aware of some fellow's work on XGBoost for kernel selection in GEMM operations. However, it required substantial feature engineering and other tweaks, which can significantly increase development time when adapting to a new GPU or firmware updates. Convolution operations are even more complex than GEMM.
>
> We remain committed to refining our work and addressing your feedback in a future revision. Thank you again for your constructive comments and for helping us improve the quality of our research.

---

### Meta-Review · Area_Chair_KQcS · 2024-12-13

**Metareview:**

This paper introduces Reformer, a lightweight, CPU-efficient deep learning model for runtime convolution kernel selection. By framing kernel selection as a sequence classification problem and leveraging the Γ-block as an alternative to transformer blocks, the authors aim to address computational and energy inefficiencies in GPU-accelerated machine learning operations. Experimental results demonstrate a 3$\times$ speedup in convolution operations on AMD MI100 GPUs compared to hand-tuned heuristics.

The strengths of the paper include a novel approach to kernel selection with practical implications for improving GPU utilization and energy efficiency. The Γ-block offers a clear direction in reducing computational overhead for CPU deployment, and the proposed ratio-preserving probability method is an interesting contribution to training robust models under noisy runtime measurements. However, significant weaknesses were raised by the reviewers, particularly in the limited evaluation scope. The results are restricted to AMD GPUs and the MIOpen library, with no comparison to state-of-the-art techniques like AutoTVM. Additionally, the authors did not provide ablation studies or discuss the scalability of the approach in-depth.

The authors submitted a brief response to the reviews but did not meaningfully engage with the reviewers' feedback or provide revisions, leaving many critical questions unanswered.

While the paper has potential, these limitations, coupled with insufficient evidence of generalizability and a lack of comparisons to relevant baselines, lead to a recommendation to reject.

**Additional Comments On Reviewer Discussion:**

The reviewer discussion highlighted several key points: (1) Multiple reviewers emphasized the need for evaluation across different GPU architectures and libraries. The authors acknowledged this but deferred addressing it to a future revision. (2) Reviewers noted the lack of comparison with other kernel optimization methods (e.g., AutoTVM, XGBoost). This was not addressed in the rebuttal. (3) Concerns about limited evaluation datasets and the absence of detailed scalability analysis were raised. While the authors suggested that their methodology could extend to other operations, they provided no supporting results or evidence. (4) Some reviewers questioned the necessity of the Γ-block and noted the incremental improvement over existing techniques. The authors did not provide compelling arguments in their rebuttal to justify this design choice.

Overall, while the reviewers recognized the potential application of the proposed approach, the limited scope, lack of engagement during the rebuttal, and unaddressed concerns about novelty, evaluation, and generalizability weighed heavily towards recommending rejection for the paper.

---

### Decision · Program_Chairs · 2025-01-22

Reject